# The Relationship between Emotional Regulation and School Burnout: Structural Equation Model According to Dedication to Tutoring

**DOI:** 10.3390/ijerph16234703

**Published:** 2019-11-26

**Authors:** Ramón Chacón-Cuberos, Asunción Martínez-Martínez, Marina García-Garnica, María Dolores Pistón-Rodríguez, Jorge Expósito-López

**Affiliations:** Department Methods of Research, Faculty of Education. University of Granada, 10871 Granada, Spain; rchacon@ugr.es (R.C.-C.); asuncionmm@ugr.es (A.M.-M.); lolapiston@ugr.es (M.D.P.-R.); jorgeel@ugr.es (J.E.-L.)

**Keywords:** schoolchildren, school burnout, emotional regulation, tutoring

## Abstract

School burnout constitutes a current phenomenon which generates diverse negative consequences in the personal and academic lives of students. Given this situation, it is necessary to develop actions that permit us to regulate this harmful mental state and that are administered from within the school context. A descriptive and cross-sectional study is presented that pursues the objective of examining a structural equation model which brings together burnout and emotional regulation. The model assumes that students receive tutoring at school in order to tackle these types of problems. For this, the sample constituted a total of 569 students from the province of Granada (men = 52.3% (n = 298); women = 47.7% (n = 271)). Mean age was reported as 10.39 ± 0.95 years and the School Burnout Inventory (BMI) and the Emotional Regulation Scale were utilized as the principal instruments. As main findings it was observed that students who received one hour of weekly tutoring showed a positive relationship between expressive suppression as a strategy of emotional regulation, cynicism, and exhaustion as consequences of school burnout. In the same way, a direct association existed between burnout-related exhaustion and cognitive repair. Given that significant relationships could not be observed between these variables in students who do not receive tutoring, higher use of emotional regulation was confirmed amongst tutored students when faced with this negative mental state.

## 1. Introduction

Burnout has traditionally been treated according to economic and social factors, without considering the underlying aspects that relate to individuals themselves [1,2]. Nevertheless, scientific studies have directly linked burnout to personal components, specifically, to the individual’s ability to regulate their emotions [3,4]. In this way, this capacity will clearly determine the way in which individuals face certain situations.

These emotional competencies must be integrated into educational processes, not only as a fundamental element in the development of the individual’s personality but as a way of optimizing the process itself and achieving a high level of training efficiency that avoids elements that can distort personality or cause burnout [5,6]. In this way, this integration must be facilitated from contexts such as guidance and the personal tutoring of students [7,8].

In this sense, burnout is an individual psychosocial state characterized by mental and emotional exhaustion, cynicism or dehumanization, lack of motivation, and the sensation of low personal achievement. This is produced as a result of overwhelming and chronic work stress [9,10,11,12]. It is provoked by individual and contextual factors such as excessive pressure, conflicts in the work environment, and the absence of rewards or emotional recognition [13,14]. In any case, as summarized by the latest International Classification of Diseases—ICD-11 [15], burnout refers to a form of work stress that is not appropriately managed.

When we link burnout to the school environment we transfer the factors that characterize this syndrome and have been identified within the work context [16], where the teaching sector represents the most researched group [17], to an educational setting within which students are directly affected [18,19]. Burnout is a useful construct in the school environment, after all it “is the context within which students work” [20] (p. 12). This is to say, it is where they carry out their basic activities (attend classes, study, complete activities, sit exams, etc.). Thus, school burnout is defined as the absence of wellbeing in the educational context. It is characterized by the presence of a feeling of exhaustion when faced with study demands, a cynical and detached attitude towards studying, and a feeling of incompetence and inadequacy as a student [21,22].

Further, this construct is multidimensional in nature [23] and incorporates three conceptually related components [20,24]: (1) school exhaustion which is understood as feelings of tension, tiredness, and chronic fatigue due to excessive school work; (2) cynicism towards school which is manifested as distant, indifferent, and disinterested behavior towards school work in general; and (3) inadequacy or incompetence at school which is considered as the feeling of lacking efficacy and achievements, or of decreased competence in school work and in school in general.

The multidimensional nature of burnout has been demonstrated with the initial study developed by Maslach, Jackson, and Leiter [25], who relied on the General Theory of Stress (GTS) to explain burnout as a negative feeling generated by interpersonal stressors associated with work. In this line, these authors developed the Multidimensional Theory of Burnout (MTB) [26], in which this concept is defined as a lack of motivation and energy associated with high levels of apathy and which is especially related to work contexts. Its main dimensions as depersonalization, the decrease of the personal achievement, and the emotional exhaustion [26]. Therefore, various studies have used this scale in the labor and educational context, making the appropriate adaptations of the scale based on the particularities of each population [27,28]. This basis leads to the definition of three basic dimensions in its adaptation to the school context carried out by Salmela-Aro, Kiuru, Leskinen, Nurmi [24] and on which the present study is supported. These are the academic exhaustion that replaced the emotional exhaustion and the cynicism that replaced the depersonalization dimension. Finally, the last dimension is inadequacy with the school, and all of them allow to facilitate the diagnosis of this psychosocial factor in school populations.

Research has confirmed that suffering burnout has important consequences for the mental (frustration, dissatisfaction, guilt, anxiety, or depression) and physical health (psycho-somatic problems, cardiovascular diseases, or musculoskeletal disorders) of individuals [29,30]. Concretely, in the school environment burnout is a high risk factor. It paves the way to demotivation, absenteeism, and school dropout [21]; whilst also negatively influencing academic engagement and school performance [31]. In contrast, school wellbeing and the absence of burnout favor successful educational trajectories [32].

On the other hand, in relation to emotional regulation (ER), one of the first concepts of the term, that remains valid in the present day, is referred to “a series of intrinsic and extrinsic processes that are responsible for monitoring, evaluating and modifying emotional responses, especially in their temporal elements and intensity, in order to achieve personal goals” [33] (pp. 27,28). It has also been stated by some authors [34] that the definition of the term has been revised and enriched as a result of research conducted into emotional regulation. Thus, emotional regulation is the process that enables individuals to identify, modulate and express their emotions whilst they interact with other individuals or situations. The purpose of this is to reach goals and objectives, to adapt to the context, or achieve personal and social wellbeing [17,35,36]. This will assist in the avoidance of burnout and forms the object of the present study.

In general, interest in emotional regulation has increased over recent years given that it is linked to wellbeing, health, personal development, performance, empathy, and interpersonal relationships [35,37]. Definitively, the management or control of one’s own emotions is fundamental for emotional wellbeing and healthy human functioning [17].

Specifically during childhood and adolescence, appropriate emotional control that enables inappropriate impulses to be inhibited, redirects behavior in a constructive way, and adapts to the situation, is key to success in interpersonal relationships, overcoming complex situations, goal achievement and, essentially, psychological and emotional wellbeing [34].

Research studies that relate these constructs have mostly been conducted within the work setting, particularly within those professions that typically present higher levels of burnout or a greater risk of suffering from it. Concretely, the teaching body represents the group most investigated in the burnout literature given that it presents record levels of exhaustion and fatigue [17].

These investigations have established that emotional regulation plays a central role in burnout [38] and that it is important to reflect upon the multiple strategies employed in the process of self-regulation that are directed towards controlling and encouraging emotional wellbeing [17]. In this sense, individuals who demonstrate a greater capacity of emotional regulation (ERA) remain open to both positive and negative emotions. They are able to recognize the value of emotions as a function of the situation and commit to them, or move away from them, depending on their usefulness. Further, they have the ability to discern which strategy is most appropriate when dealing with a specific emotion [29].

As a result, emotional regulation has the opposite effect on burnout [39]. Definitively, emotional regulation and, specifically, the emotional regulation ability (ERA), are key to burnout [29,40,41]. In this sense, different authors [17,29,39], moved by the results of their work, show the relevance of pushing for intervention programs targeting the development of social and emotional skills and emotional regulation strategies.

Further, “results demonstrate that school burnout is positively associated with emotional deregulation” in adolescents [42] (p. 18), specifically in an “emerging adults” sample (*M* = 19) in the southeastern United States. In fact, in a study carried out with Spanish secondary school students between 12 and 18 years, a significant negative relationship has been found to exist between the emotional intelligence sub-scales, between them, emotional regulation, and school burnout [31].

The review of these studies makes it possible to establish the importance of also promoting the development of programs and activities within the educational setting. These actions should be directed towards working on emotional skills in the classroom, emphasizing the capacity to regulate emotions in different situations by putting strategies into action that are adapted to these very emotions. Developing the ERA of students will decrease the likelihood of them suffering school burnout, whilst at the same time favoring their wellbeing and academic success. These interventions should be proposed from a multidisciplinary and transversal standpoint, an approach that is typically taken in school tutoring [7,43].

This new model of tutoring [8,44] that include teachers, families, and students as active agents not only at school, but also in the family and social context, is usually called Tutorial Action [39,45,46]. However, above all, it transcends the traditional contents of academic guidance, focused on the orientation for studies; and includes guidance for development of the student’s personality, on topics such as self-esteem, self-concept, and emotional intelligence; for improvement of learning [47,48], on the use of appropriate learning strategies and study techniques; and for development of social and professional skills, developing skills of communication, social responsibility, and sustainable knowledge transfer [49].

To this end, the present study pursues the following objectives:To develop a structural equation model that enables associations between school burnout, emotional regulation, and the age of schoolchildren to be defined.To compare existing associations between the variables that compose the path model according to treatment during classroom tutoring through a multigroup analysis.

## 2. Materials and Methods

### 2.1. Design and Participants

The present study describes a nonexperimental ex post-facto study design that is descriptive and cross-sectional in nature and conducted amongst schoolchildren from the province of Granada (Spain). A single measurement was conducted within a single group, composed of a total sample of 569 schoolchildren belonging to eight public centers, which are located in different areas of the province of Granada with diverse socio-economic contexts. Participants were attending the second or third year of primary education, with age being reported between 8 and 13 years (*M* = 10.39; *SD* = 0.95). The sample consisted of 52.3% (n = 298) males and 47.7% (n = 271) females. The study sample was recruited using a multistage process. Educational centers were selected intentionally as they already had a pre-existing collaborative relationship with the research team. They were informed of the study aims, procedures, and instruments for collecting information and handling data, and a sample of volunteers from the School Council were invited to participate. The collaboration of natural-class groups was requested, whilst adhering to legal requirements around the participation of schoolchildren in research. The following selection criteria was determined: (a) individuals who were enrolled in the fourth, fifth, or sixth grade of primary education during the 2018/2019 school year at selected schools in the province of Granada. The following exclusion criteria was established: (b) individuals who did not possess established legal authorizations or possessed any impediment which impeded them from completing any of the instruments proposed for data collection.

### 2.2. Instruments

The following instruments were employed for the measurement of the proposed variables:School Burnout Inventory, developed in 2009 [24] and validated in Spanish in 2013 [5]. This scale is composed of nine items (e.g., “1. I feel that I am not able to complete all of my school work”) which are rated along a Likert scale with five response options (1 = completely disagree; 5 = completely agree). This questionnaire groups school burnout according to three dimensions: exhaustion (items 1, 4, 7, and 9), cynicism (items 2, 5, and 6), and inefficacy (items 3 and 8). This instrument presents adequate internal consistency with this specific sample, with a Cronbach alpha of *α* = 0.746Emotional Regulation Questionnaire, developed in 2003 [50] and validated in Spanish in 2016 [6]. This scale is composed of 10 items (e.g., “1. I keep my emotions to myself”), which are rated along a Likert scale with seven response options (1 = completely disagree; 7 = completely agree). This questionnaire groups emotional regulation into two dimensions: (1) expressive suppression, which is associated with the nature of the emotional response, implying a decrease in the expressive behavior of the emotion that is being felt (items 1, 2, 3, and 4), and (2) cognitive reevaluation, which is related to the way of interpreting cognitively the information that a subject experiences, modulating their emotional meaning, and also with the ability he has to face it (items 5, 6, 7, 8, and 9). The instrument revealed acceptable, although relatively low, internal consistency with this specific sample. The value of Cronbach alpha was *α* = 0.633.Self-registration sheet. An “ad-hoc” questionnaire was attached with variables that were socio-demographic in nature and related with the tutoring approach. It considered sex (male/female); age; school year (3rd/4th/5th/6th), school/institute; dedicates 1 h a week to tutoring (yes/no).

### 2.3. Procedure

The research team informed the educational centers of the study aims, measurement instruments, and how the data would be handled following its collection. Once the permissions were obtained that permit access to the research setting, the instruments were administered in the presence of a research interviewer. The purpose of this was to ensure correct application of the scales and to resolve any doubts that might emerge during data completion. Data were collected and its quality was confirmed, whilst ensuring throughout that the process conformed to the ethical principles for research defined in the Declaration of Helsinki in 1975 and later updated in Brazil in 2013 [51].

### 2.4. Data Analysis

Data analysis was conducted using the software IBM SPSS^®^ for Windows, version 23.0. Frequencies and means were employed for the basic descriptive analysis. Likewise, Cronbach’s coefficient was utilized in order to determine internal consistency of the instruments, fixing the reliability index at 95%. Multi-group analysis through structural equations (SEM) was carried out using the software IBM AMOS^®^ version 23.0 (International Business Machines Corporation, Armonk, NY, USA). SEM was utilized in order to determine the relationships between the variables that constitute the theoretical model (Figure 1) for both groups (schoolchildren enrolled in formative processes that involve a commitment to tutoring for their emotional development and schoolchildren enrolled in training processes without a commitment to tutoring specifically targeting these aspects.

The SEM was constituted by six exogenous variables. The model provided explanations of these variables through observed associations. Observed variables are those that present an error term that is represented using a circle, whilst latent variables do not present error terms and employ bidirectional arrows. In this way, the latent variables were Burnout-Exhaustion (B-EXH), Burnout-Cynicism (B-CIN), and Burnout-Inefficacy (B-INA). On the other hand, the observable variables were Emotional Regulation-Expressive Suppression (RE-ES), Emotional Regulation-Cognitive Repair (RE-CR), and Age (Figure 1).

Bidirectional arrows show the associations between the latent variables (covariances), whilst the unidirectional arrows show the associations between the observable variables and their associated error terms, which are interpreted as multivariate regression coefficients. Prediction errors relate the observable and endogenous variables to the model. Likewise, the maximum likelihood method (ML) was employed to estimate the associations between variables as it is consistent and invariable to scale type.

With the purpose of determining compatibility of the SEM with the empirical information obtained, different indices were employed to determine fit of the theoretical model. Nonsignificant values should be obtained for the *p*-value, however, other fit indices should be employed as this statistic is highly sensitive to the effects of sample size [52]. Amongst these other indices, the Comparative Fit Index (CFI), Incremental Fit Index (IFI), Normalized Fit Index (NFI), and the Tucker–Lewis Index (TLI) were utilized. For these, values higher than 0.90 should be obtained in order to present an acceptable fit and values higher than 0.95 present excellent fit indices. Further, the Root Mean Square Error Approximation (RMSEA) was used with acceptable fit being determined by values lower than 0.08 and excellent fit by values lower than 0.05.

## 3. Results

The structural model developed showed good fit indices for the multi-group analysis. The chi-squared test revealed a statistically significant value (χ^2^ = 7.389; *df* = 8; *p* = 0.495). Given the sensitivity to sample size presented by this statistic, the relevance of using other standardized fit indices is noted [52]. In this way, the NFI obtained a value of 0.976, the IFI revealed a value of 0.982, and the CFI produced a value of 0.986, all of these representing excellent fit. Likewise, the RMSEA obtained a value of 0.036, this also being excellent and demonstrating an appropriate level of adjustment of the SEM.

Table 1 and Figure 2 show the regression weights and standardized regression weights for the structural model developed within schoolchildren who dedicate one hour a week to work on questions linked to tutoring (academic, personal, and emotional). This enabled determination of the associations between school burnout, emotional regulation, and age. Statistically significant associations are shown (*p* < 0.005) at the first level of the model between the three dimensions of school burnout, these all being positive and direct, with the strength of the relationship as indicated by the regression weight being as follows: cynicism and inefficacy (*b* = 0.561), exhaustion and cynicism (*b* = 0.527), and exhaustion and inefficacy (*b* = 0.467).

Following this, associations were shown between the dimensions of school burnout and emotional regulation. The strongest association is observed for the relationship between exhaustion and expressive suppression (*p* < 0.005; *b* = 0.231), followed by the association between cynicism and expressive suppression (*p* < 0.01; *b* = 0.161) and finally, exhaustion and cognitive repair (*p* < 0.01; *b* = 0.149). Statistically significant differences were not observed at this level for any other associations.

Finally, the relationship between both dimensions of emotional regulation are shown, with statistically significant differences being revealed (*p* < 0.01; *b* = 0.165). Likewise, expressive suppression was negatively associated with age (*p* < 0.05; *b* = −0.096).

Table 2 and Figure 3 show the regression weights and standardized regression weights of the structural model developed for schoolchildren who did not have an hour each week dedicated to work on questions linked to tutoring. In this way, relationships can be determined between school burnout, emotional regulation, and age within participants who do not tackle academic, personal, or emotional questions with a tutor. Statistically significant associations (*p* < 0.005) are shown at the first level of the model between the three dimensions of school burnout. All relationships are positive and direct, with the strongest to weakest regression weights being as follows: cynicism and inefficacy (*b* = 0.509), exhaustion and inefficacy (*b* = 0.407), and exhaustion and cynicism (*b* = 0.372).

For this model, a statistically significant association was only observed between school burnout and emotional regulation, this being evident in the dimensions of inefficacy and cognitive repair (*p* < 0.05; *b* = 0.225). Age was also associated with emotional regulation, whilst statistically significant differences were also observed between expressive suppression and cognitive repair (*p* < 0.05; *b* = 0.230).

## 4. Discussion

The present research work sought to compare an explanatory model of existing relationships between burnout and emotional regulation in a sample of Spanish schoolchildren. It followed the line of research presented by some similar studies which have approached wellbeing at school and emotional education [31,39,42]. Likewise, a multi-group analysis was carried out with the purpose of identifying existing differences in the relationships between these variables, considering dedication to tutoring in the educational context. Concretely, the relevance of tutoring for ensuring the integral development of students, considering its academic, socio-affective, and cognitive dimensions [53]. To this end, the development of tutoring is presented as an essential element that encapsulates a set of actions planned by various professionals from the educational context and coordinated by the tutor. These actions may be able to help in the treatment of school burnout [39,54].

In reviewing the first level of the structural model, a positive relationship can be observed between the dimensions of burnout—exhaustion, cynicism, and inefficacy—in the two analyzed groups. The strength of association was moderate in both schoolchildren receiving dedicated tutoring time and those who were not benefiting from this process of guidance and orientation. These findings confirm that a consistent inter-relationship exists amongst the variables that form the construct of interest, providing a theoretical model of burnout that can be justified by the inter-relationship evident in the three factors that compose school burnout [55]. In fact, this state of physical and mental exhaustion, alongside a lack of motivation, is linked to negative conducts at school. Such behaviors can include lack of approval of the processes taking place in school, stress, and low perception of one’s own ability to meet educational requirements [5]. It seems to be the case that once one finds them self in this mental state an inter-relationship is produced between exhaustion and inefficacy, independent of the actions introduced to tackle it [21,56].

The second level of the model tackles the relationship between school burnout and emotional regulation. Specifically, it was observed that pupils who receive one hour of tutoring (the development of emotional competences is considered to be amongst one of the fundamental contents according to new models of tutoring attention) demonstrated a positive relationship between cynicism generated by burnout and expressive suppression (the attempt to eliminate negative thoughts), without this relationship being seen in schoolchildren who do not receive tutoring. Tutoring and psychopedagogical orientation constitute a basic means of emotional education. This is because tutoring enables specialized attention to be paid to students in their personal and social ambits. This then helps students to understand and regulate their own emotions, supporting the development of their wellbeing and the configuration of their attitudes towards school [8,43,57]. These premises cement the idea that students who are appropriately tutored will better understand their emotions and possess regulation strategies. In this way they can move to eliminate negative thoughts by putting strategies for expressive suppression into place before burnout associated cynicism even appears [29,58].

Along similar lines to that presented, it was observed that schoolchildren who receive one hour of tutoring show a positive relationship between exhaustion generated by burnout and expressive suppression, the same as that which happened with cynicism. On the other hand, a positive and direct relationship existed between burnout associated exhaustion and cognitive repair (the generation of positive thoughts when faced with adverse situations). Emotional education provided through educational orientation in tutorials will help to prevent learning and adaptation difficulties [59,60]. This, in addition to it helping to develop skills for school and social life, makes it a stand-out element in burnout prevention. This could justify the fact that students who present burnout symptoms and receive tutoring, are those who also possess more strategies for emotional regulation. Concretely, the act of tutoring constitutes a planned, systematic, and proactive process that permits the prevention of diverse risk factors and the treatment of a multitude of problems, amongst which burnout is found [7].

Definitively, tutoring describes a continuous process that enables the tutor to identify stressful agents linked to the state of burnout [61]. Tutoring actions permit groups of teaching staff to modify their practice in order to favor the generation of a positive perception of school amongst students, eliminating the negative elements and actions that lead to feelings of inefficacy and exhaustion [17,62]. Further, building on the aforementioned conclusion, the tutor will be able to provide their students with coping strategies and emotional regulation. This means the problem can be approached in a bidirectional way. In this way, the development of tutoring is understood as an educational process that pursues integral development of students [7]. It will help in the prevention and treatment of this problem which has expanded in the 21st century and is associated with other negative consequences such as poor academic performance, school anxiety, and depression [63,64].

The third level of the model reveals an association between the two dimensions of emotional regulation and age. Concretely, this suggests that schoolchildren who do not receive tutoring show a direct and strong relationship between expressive suppression and cognitive repair. Expressive suppression, whilst being an emotional regulation strategy, has fewer positive connotations than those linked with cognitive repair [6]. In this way, predominance of the latter within schoolchildren receiving tutoring is logical and explains the weak relationship between both dimensions in this group. In this sense, actions realized by the tutor will help in the construction of essential psychosocial factors and development of positive character traits which will help prevent risk behaviors [59]. Finally, with regard to age, it is highlighted that this was inversely related with expressive suppression within students who received tutoring, although the regression weight was relatively weak. Specifically, age can represent a relevant factor of emotional regulation, in the way that previous experience will help cognitive repair learning as a predominant strategy for expressive suppression [65].

Finally, it is interesting to identify some of the main limitations of the present study. Firstly, the nature of the study should be highlighted. As the study was nonexperimental it does not permit causal relationships to be established between variables. However, this type of study is highly efficient for understanding the state of the question. Another limitation resides in the sample used, as probabilistic sampling was not employed, and the sample was not representative of the population of Granada. This being said, it should be noted that the number of participants was large and the selection of natural groups in the educational context implies a degree of randomness, as has been stated in the literature [66]. On the other hand, using only self-report measures entails the risk that participants omit information or misunderstand an issue. As a final limitation, the instruments administered can be highlighted. Whilst they were validated within schoolchildren and adolescents, high internal consistency was not obtained. Thus, the need for future studies to adapt the scales employed here is proposed.

## 5. Conclusions

As the main findings, the structural equation model proposed revealed that schoolchildren who receive one hour a week of tutoring show a positive relationship between expressive suppression as a strategy of emotional regulation, and cynicism and exhaustion as consequences of school burnout. In the same way, a direct relationship existed between exhaustion linked to burnout and cognitive repair. Given that significant relationships could not be observed between these variables for schoolchildren who were not receiving tutoring, a greater use of emotional regulation is noted within tutored schoolchildren when faced with this negative mental state. It makes sense to point out that tutoring is an educational dimension that is considered to be anachronistic, poorly founded, and rarely done in a coordinated manner. For this reason, a conceptualization and transformation of its practice is required [7]. Training as a way to develop personality, in which improvement and development of emotional competences should be included. In the same way, tutorials involve a large number of actions coordinated by the tutor, in which diverse agents from the educational community are incorporated in order to implement a planned and structured process, based on providing help and support to students to strengthen their integral development [67]. This must be in addition to tutoring being worked in an integrated or infusive way [68]. The actions performed by the tutor make up new tasks that are embedded within a new contemporary teaching profile. These tasks are more focused on paying attention to the student rather than on the subject matter or content being studied. Guidance, tutoring, or mentoring can be considered the ideal educational setting from which to tackle the new educational matters of personal development and emotion management. Further, tutoring offers a more appropriate strategy to transversally introduce these aspects into other school materials and environments. Definitively, all of these aspects make up specific actions upon which new models of educational attention can be developed. This is supported by the present work which specifically demonstrates the link between the management of emotions at different educational levels and school burnout prevention. Generally, the findings allow new intervention variables to be delimited and considered for the improvement and development of higher quality education.

## Figures and Tables

**Figure 1 ijerph-16-04703-f001:**
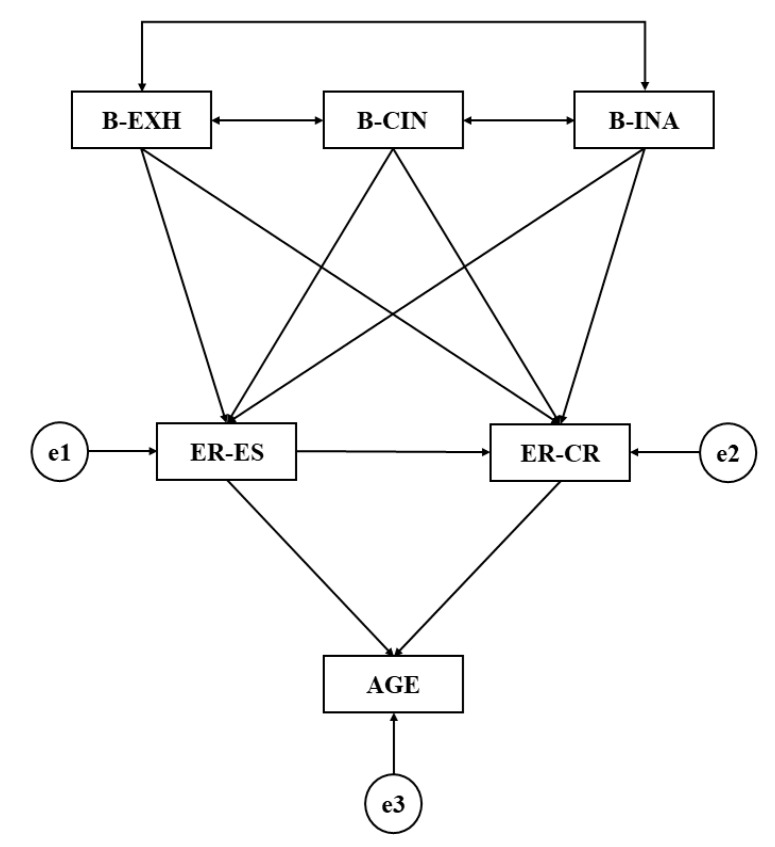
Theoretical model. Source: elaborated by the authors. Note 1: B-EXH, Burnout-Exhaustion; B-CIN, Burnout-Cynicism; B-INA, Burnout-Inadequacy; ER-ES, Emotional Regulation-Expressive Suppression; ER-CR, Emotional Regulation-Cognitive Repair.

**Figure 2 ijerph-16-04703-f002:**
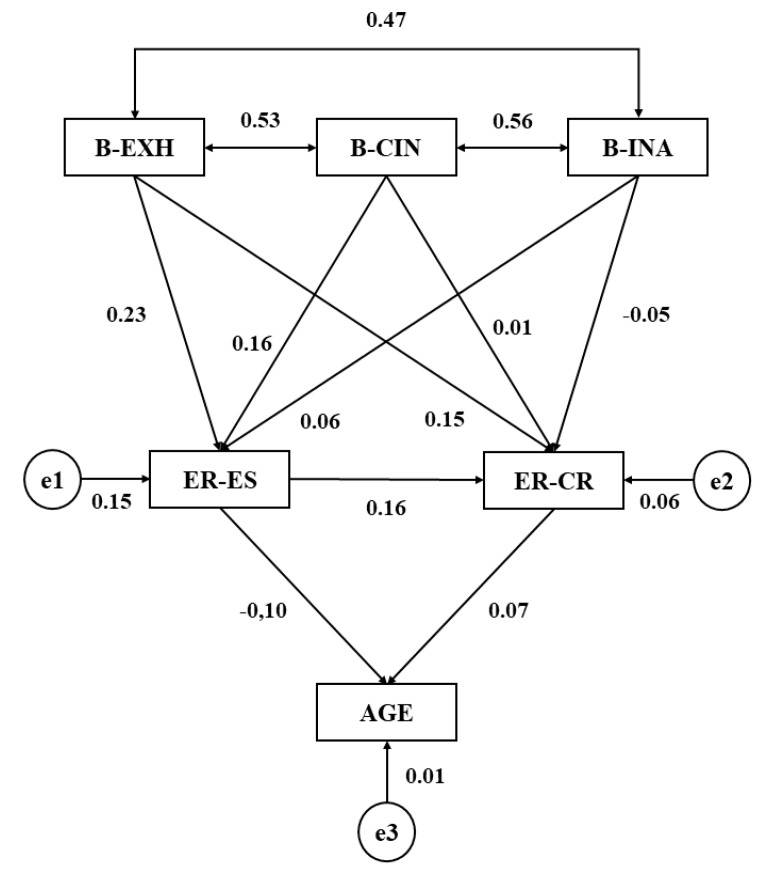
Structural equation model for schoolchildren receiving dedicated tutoring time. Source: Elaborated by the authors. Note 1: B-EXH, Burnout-Exhaustion; B-CIN, Burnout-Cynicism; B-INA, Burnout-Inefficacy; ER-ES, Emotional Regulation-Expressive Suppression; ER-CR, Emotional Regulation-Cognitive Repair.

**Figure 3 ijerph-16-04703-f003:**
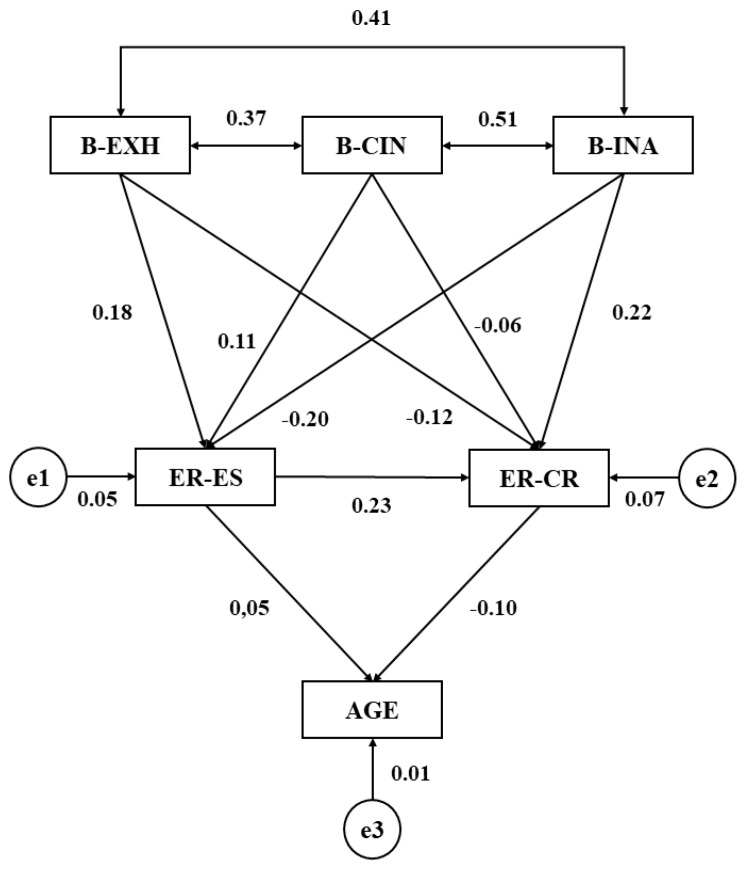
Structural equation model for schoolchildren not receiving dedicated tutoring time. Source: Elaborated by the authors. Note 1: B-EXH, Burnout-Exhaustion; B-CIN, Burnout-Cynicism; B-INA, Burnout-Inefficacy; ER-ES, Emotional Regulation-Expressive Suppression; ER-CR, Emotional Regulation-Cognitive Repair.

**Table 1 ijerph-16-04703-t001:** Regression weights in schoolchildren receiving dedicated tutoring time.

Relationships between Variables			RW				SRW
			EST	EE	CR	P	EST
RE-ES	←	B-EXH	0.366	0.084	4.345	***	0.231
RE-ES	←	B-CIN	0.203	0.072	2.839	**	0.161
RE-ES	←	B-INA	0.073	0.067	1.091	0.275	0.059
RE-CR	←	B-EXH	0.194	0.074	2.612	**	0.149
RE-CR	←	B-CIN	0.014	0.062	0.222	0.825	0.013
RE-CR	←	B-INA	−0.050	0.057	−0.874	0.382	−0.050
RE-CR	←	RE-ES	0.135	0.041	3.312	**	0.165
Age	←	RE-ES	−0.061	0.030	−1.998	*	−0.096
Age	←	RE-CR	0.057	0.037	1.528	0.126	0.074
B-INA	⇄	B-EXH	0.545	0.061	8.930	***	0.467
B-EXH	⇄	B-CIN	0.594	0.060	9.840	***	0.527
B-INA	⇄	B-CIN	0.823	0.080	10.325	***	0.561

Note 1: EST, Estimation; EE, Estimation Error; CR, Critical Ratio; RW, Regression Weights; SRW, Standardized Regression Weights. Note 2: B-EXH, Burnout-Exhaustion; B-CIN, Burnout-Cynicism; B-INA, Burnout-Inadequacy; ER-ES, Emotional Regulation-Expressive Suppression; ER-CR, Emotional Regulation-Cognitive Repair. Note 3: ***, Statistically significant differences at level *p* < 0.005; **, Statistically significant differences at level *p* < 0.01; *, Statistically significant differences at level *p* < 0.05.

**Table 2 ijerph-16-04703-t002:** Regression weights for schoolchildren not receiving dedicated tutoring time.

Relationships between Variables	RW				SRW
	EST	EE	CR	P	EST
RE-ES←B-EXH	0.260	0.145	1.787	0.074	0.178
RE-ES←B-CIN	0.138	0.139	0.995	0.320	0.105
RE-ES←B-INA	−0.234	0.128	−1.833	0.067	−0.197
RE-CR←B-EXH	−0.135	0.111	−1.218	0.223	−0.121
RE-CR←B-CIN	−0.056	0.105	−0.536	0.592	−0.056
RE-CR←B-INA	0.204	0.097	2.095	*	0.225
RE-CR←RE-ES	0.176	0.068	2.566	*	0.230
Age←RE-ES	0.030	0.060	0.500	0.617	0.046
Age←RE-CR	−0.081	0.079	−1.031	0.303	−0.095
B-INA⇄B-EXH	0.476	0.115	4.139	***	0.407
B-EXH⇄B-CIN	0.395	0.103	3.828	***	0.372
B-INA⇄B-CIN	0.663	0.133	4.984	***	0.509

Note 1: EST, Estimation; EE, Estimation Error; CR, Critical Ratio; RW, Regression Weights; SRW., Standardized Regression Weights. Note 2: B-EXH, Burnout-Exhaustion; B-CIN, Burnout-Cynicism; B-INA, Burnout-Inadequacy; ER-ES, Emotional Regulation-Expressive Suppression; ER-CR, Emotional Regulation-Cognitive Repair. Note 3: ***, Statistically significant differences at level *p* < 0.005; **, Statistically significant differences at level *p* < 0.01; *, Statistically significant differences at level *p* < 0.05.

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
