# Peer review of "The Relationship between Emotional Regulation and School Burnout: Structural Equation Model According to Dedication to Tutoring"

_ijerph, 2019, doi:10.3390/ijerph16234703_

Round 1

Reviewer 1 Report

This article provides an interesting model describing the relationships between burnout and emotional regulation among a sample of Spanish school-aged children. The authors provide a thorough description of their methods and analyses, and the manuscript generally appears to be well crafted.

I have several comments for further improving the manuscript. These include:

The current International Classification of Diseases is referred to as the ICD-11.

Additional information or examples to clarify the exclusionary criteria for participants would be helpful. Possessing “any impediment” is not clear or specific.

A brief description of the emotion regulation variables (expressive suppression and cognitive repair) would be helpful.

The information presented on tutoring seems rather scant. Further detail about the tutoring process would also be helpful.

Author Response

We would like to express our gratitude for the time taken to review this manuscript and for the comments made, which we believe to be critical for producing rigorous and quality research. We have detailed below the changes made in the original article: “The Relationship between Emotional Regulation and School Burnout: Structural Equation Model according to Dedication to Tutoring” (ijerph-635721).

Modifications have been made in the original manuscript following the reviewers’ comments. For each modification, we have written: the original comment as written by the reviewer in addition to the page and line number; and the change made in response to that comment. Changes have been made using the tool “Track changes” enabling editor and reviewers to identify modifications easily.

MODIFICATIONS

Comment 1:

The current International Classification of Diseases is referred to as the ICD-11.

Response 1:

Thank you for this suggestion for improvement. This abbreviation has been modified in line 46.

Comment 2:

Additional information or examples to clarify the exclusionary criteria for participants would be helpful. Possessing “any impediment” is not clear or specific.

Response 2:

Thank you for this suggestion for improvement. The inclusion and exclusion criteria are specified in lines 212-216:

Be a student of the fourth, fifth or sixth grade of primary education during the 2018/2019 school year at schools in the province of Granada that collaborates with the research team. Have an authorization to participate in the research. Have no impediment.

When we say that the student has no impediment, we are referring that student does not suffer from some difficulty or disorder that makes it difficult for them to read or complete the questions when responding to the self-registration template.

Comment 3:

A brief description of the emotion regulation variables (expressive suppression and cognitive reevaluation) would be helpful.

Response 3:

Thank you for this suggestion for improvement. We have introduced a footnote on page 4 with a brief description of the emotion regulation variables:

  1Expressive suppression is associated with the nature of the emotional response, implying a decrease in the expressive behavior of the emotion that is being felt.

  2Cognitive reevaluation is related to the way of interpreting cognitively the information that a subject experiences, modulating their emotional meaning, and also with the ability he has to face it.

Comment 4:

The information presented on tutoring seems rather scant. Further detail about the tutoring process would also be helpful.

Response 4:

Thank you for this suggestion for improvement. We provide more details about the tutoring process in lines 145-152:

“This new model of tutoring [8,41] that include teachers, families and students as active agents not only at school, but also in the family and social context, is usually called Tutorial Action [42,43,36]. But above all, it transcends the traditional contents of academic guidance, focused on the orientation for studies; and includes guidance for development of the student's personality, on topics such as self-esteem, self-concept and emotional intelligence; for improvement of learning [44,45], on the use of appropriate learning strategies and study techniques; and for development of social and professional skills, developing skills of communication, social responsibility and sustainable knowledge transfer [46].”

Reviewer 2 Report

This study aims to examine the relationship between school burnout and emotional regulation using a structural equation model.

The article is well structured and the results section is well presented. However, there are still some items that need the author’s' attention, as follows:

Comment 1. Research studies about burnout have been focused mainly on teachers. This idea should be mentioned before, specifically when the authors talk about burnout and school environment (line 48). This allows to highlight the novel of this research analyzing the burnout associated with students (school burnout). In addition, the theoretical framework should develop in some terms the variable Tutoring.

Comment 2. A better literature review about the multi-dimensional nature of the school burnout is required. Please, indicate if the three conceptually related components explained are based on some theory or model (lines 56-57).

Comment 3. Please, check the explanation of the abbreviature of ERA (lines 93 and 99), I think is more appropriate emotion-regulation ability. In addition, the abbreviature of ER appears the first time in line 99, however, this concept is used previously. Please, present this abbreviature before.

Comment 4. I recommend to indicate the nationality and age range of the participants in the studies mentioned, for example in lines 103-104 when the results are indicated in adolescents specify the nationality, in this way the reader can know in which countries have developed more this type of studies.

Comment 5. With regard to the method, the sample’s socio-economic context does not appear. In addition, regarding instruments, the authors should explain in more detail the different dimensions that make up the measures and specify if the internal consistency mentioned in each instrument is induced from previous studies or it is the result obtained in this research. I recommend indicating the internal consistency of each instrument with this specific sample.

Comment 6. There is some statistical expression that should appear on italic in the manuscript and tables (for instance: p, M, etc.). Regarding tables, please, translate Edad to Age (see tables 1 and 2) and revise the abbreviature B-INA presented in the note 2.

Comment 7. In the discussion section more limitations about using only self-report measures and the students’ perspective should be mentioned (e.g. to collect information from different sources and taking into account not only the students’ perception but also the information offered by parents and teachers).

Author Response

Dear the editor and reviewer,

We would like to express our gratitude for the time taken to review this manuscript and for the comments made, which we believe to be critical for producing rigorous and quality research. We have detailed below the changes made in the original article: “The Relationship between Emotional Regulation and School Burnout: Structural Equation Model according to Dedication to Tutoring” (ijerph-635721).

Modifications have been made in the original manuscript following the reviewers’ comments. For each modification we have written: the original comment as written by the reviewer in addition to the page and line number; and the change made in response to that comment. Changes have been made using the tool “Track changes” enabling editor and reviewers to identify modifications easily.

Comment 1.

Research studies about burnout have been focused mainly on teachers. This idea should be mentioned before, specifically when the authors talk about burnout and school environment (line 48). This allows to highlight the novel of this research analyzing the burnout associated with students (school burnout). In additional, the theoretical framework should develop in some terms the variable Tutoring.

Response 1:

Thank you for this suggestion for improvement. We have taken your consideration into account and incorporated it into the lines 48-51: “When we link burnout to the school environment we transfer the factors that characterise this syndrome and have been identified within the work context, where the teaching sector represents the most researched group (Talbot & Mercer, 2018), to an educational setting within which students are directly affected”.  We provide more details about the tutoring process in lines 145-152:

“This new model of tutoring [8,41] that include teachers, families and students as active agents not only at school, but also in the family and social context, is usually called Tutorial Action [42,43,36]. But above all, it transcends the traditional contents of academic guidance, focused on the orientation for studies; and includes guidance for development of the student's personality, on topics such as self-esteem, self-concept and emotional intelligence; for improvement of learning [44,45], on the use of appropriate learning strategies and study techniques; and for development of social and professional skills, developing skills of communication, social responsibility and sustainable knowledge transfer [46].”

Comment 2.

A better literature review about the multi-dimensional nature of the school burnout is required. Please, indicate if the three conceptually related components explained are based on some theory or model (lines 56-57).

Response 2:

Thank you for this suggestion for improvement. The multidimensional nature of burnout has been explained as well as the theoretical model on which the three dimensions used are based (line 57-76).

“Further, this construct is multi-dimensional in nature [19] and incorporates three conceptually related components [16,20]: 1. School exhaustion which is understood as feelings of tension, tiredness and chronic fatigue due to excessive school work; 2. Cynicism towards school which is manifested as distant, indifferent and disinterested behaviour towards school work in general; and 3. Inadequacy or incompetence at school which is considered as the feeling of lacking efficacy and achievements, or of decreased competence in school work and in school in general. 

The multidimensional nature of burnout has been demonstrated with the initial study developed by Maslach, Jackson, & Leiter [21], who relied on the General Theory of Stress (GTS) to explain burnout as a negative feeling generated by interpersonal stressors associated with work. In this line, these authors developed the Multidimensional Theory of Burnout (MTB) [22], in which this concept is defined as a lack of motivation and energy associated with high levels of apathy and which is especially related to work contexts. Its main dimensions as depersonalization, the decrease of the personal achievement and the emotional exhaustion [22]. Therefore, various studies have used this scale in the labour and educational context, making the appropriate adaptations of the scale based on the particularities of each population [23,24]. This basis leads to the definition of three basic dimensions in its adaptation to the school context carried out by Salmela-Aro, Kiuru, Leskinen, Nurmi [20] and on which the present study is supported. These are the academic exhaustion that replaced the emotional exhaustion and the cynicism that replaced the depersonalization dimension. Finally, the last dimension is inadequacy with the school, and all of them allow to facilitate the diagnosis of this psychosocial factor in school populations.”

Comment 3.

Please, check the explanation of the abbreviature of ERA (lines 93 and 99), I think is more appropriate emotion-regulation ability. In addition, the abbreviature of ER appears the first time in line 99, however this concept is used previously. Please, present this abbreviature before.

Response3.

Thank you for this suggestion. We change “capacity for emotional regulation” to “emotional regulation ability” (line 128). It was due to a mistake in the translation process. In addition, we present the abbreviature of ER before, in line 84.

Comment 4.

I recommend to indicate the nationality and age range of the participants in the studies mentioned, for example in lines 103-104 when the results are indicated in adolescents specify the nationality, in this way the reader can know in which countries have developed more this type of studies.

Response 4.

Thank you for this suggestion for improvement. Indicate the nationality and age range of the participants in the studies mentioned in the lines 132-136:

Further, “results demonstrate that school burnout is positively associated with emotional deregulation” in adolescents [39] (p. 18), specifically in a "emerging adults" sample (n= 19) in the southeastern United States. In fact, in a study carried out with Spanish secondary school students between 12 and 18 years, a significant negative relationship has been found to exist between the emotional intelligence sub-scales, between them emotional regulation, and school burnout [27].

Comment 5.

With regard to the method, the sample’s socio-economic context does not appear. In addition, regarding instruments, the authors should explain in more detail the different dimensions that make up the measures and specify if the internal consistency mentioned in each instrument is induced from previous studies or it is the result obtained in this research. I recommend to indicate the internal consistency of each instrument with this specific sample.

Response 5.

Thank you for this suggestion for improvement. With regard to the method, we mention the sample’s socio-economic context in the line 203-204. Regarding instruments, we mention the internal consistency in each instrument as the result obtained in this research. To clarify we have added in each case the text “with this specific sample” (Line 224 and 230). We have not specified the internal consistency of each instrument in the original study because we believe it can be easily consulted according to the references cited

Comment 6.

There is some statistical expression that should appear on italic in the manuscript and tables (for instance: p, M, etc.). Regarding tables, please, translate Edad to Age (see tables 1 and 2) and revise the abbreviature B-INA presented in the note 2.

Response 6.

Thank you for this suggestion for improvement. We have changed all statistical expressions to italics in the manuscript and tables. Regarding tables, we translate Edad to Age in the tables 1 and 2 and we change the abbreviature B.INA to B-INA in figures 1 and 2 and in the note 2 in tables 1 and 2.

Comment 7.

In the discussion section more limitations about using only self-report measures and the students’ perspective should be mentioned (e.g. to collect information from different sources and taking into account not only the students’ perception but also the information offered by parents and teachers).

Response 7.

Thank you for this suggestion for improvement. We add the limitation about using only self-report measures in the line 455-456: “On the other hand, using only self-report measures entails the risk that participants omit information or misunderstand an issue”.In relation to collecting information from different sources (families, teachers), although it would have been possible, in this study in particular it was decided to include only the perception of the students because two internal psychosocial factors of the individual are evaluated

Round 2

Reviewer 2 Report

Thank you very much for your revisions.

All my comments have been successfully resolved increasing the research quality of this manuscript.

Author Response

Thank you.